# AI-HEAT: A Clinical Decision Support System for pediatrics febrile conditions

Abraham Bautista-Castillo
*Computational Biomedicine Lab*
*University of Houston*
Houston, TX, USA

Rocio A. Padilla-Medina
*Computational Biomedicine Lab*
*University of Houston*
Houston, TX, USA

Jessica Nguyen
*Division of Rheumatology*
*Baylor College of Medicine*
Houston, TX, USA

Chisato Shimizu
*Department of Pediatrics*
*Rady Children's Hospital, San Diego*
*University of California San Diego*
La Jolla, CA, USA

Adriana H. Tremoulet
*Department of Pediatrics*
*Rady Children's Hospital, San Diego*
*University of California San Diego*
La Jolla, CA, USA

Jane C. Burns
*Department of Pediatrics*
*Rady Children's Hospital, San Diego*
*University of California San Diego*
La Jolla, CA, USA

Ananth V. Annapragada
*Department of Radiology*
*Baylor College of Medicine*
*Texas Children's Hospital*
Houston, TX, USA

Tiphanie P. Vogel
*Division of Rheumatology*
*Baylor College of Medicine*
*Texas Children's Hospital*
Houston, TX, USA

Ioannis A. Kakadiaris
*Computational Biomedicine Lab*
*University of Houston*
Houston, TX, USA
ikakadia@central.uh.edu

*Abstract*—In recent years, several diagnostic challenges have developed due to the COVID-19 pandemic, including the post-infectious sequelae multisystem inflammatory syndrome in children (MIS-C). This syndrome shares several clinical features with other entities, such as Kawasaki disease (KD) and endemic typhus, among other febrile diseases. Endemic typhus, or murine typhus, is an acute infection treated much differently than MIS-C and KD. Early diagnosis and appropriate treatment are crucial to a favorable outcome for patients with these disorders. To address these challenges, a Clinical Decision Support System (CDSS) designed to support the decision-making of medical teams can be implemented to differentiate between these disorders. We developed and evaluated a CDSS based on a Triplet Loss Siamese Network to distinguish between patients presenting with clinically similar febrile illnesses, KD, MIS-C, or typhus. We used eight clinical and laboratory features typically available within six hours of presentation. The performance assessment for AI-HEAT, Logistic Regression, Support Vector Machine, XGBoost, and the TabPFN machine learning models was performed by computing Balanced Accuracy. AI-HEAT is a CDSS capable of obtaining performance similar to a state-of-the-art Transformer-type deep learning model such as TabPFN, with advantages such as being almost a thousand times smaller.

*Index Terms*—Artificial Intelligence, Clinical Decision Support System, Deep Learning, Endemic Typhus, Kawasaki, MIS-C

This work was partly supported by NIH grant number R33HD105593. Abraham Bautista-Castillo is also supported by the National Council of Science and Technology of Mexico, scholarship number 739528. Icons and diagrams shown in this work were obtained and created through free licenses from Icons8 [1] and Draw.io [2], respectively. Any opinions, findings, conclusions, or recommendations expressed in this material are those of the authors. They do not necessarily reflect the views of the NIH, other funders, the position, or the policy of the Government, and no official endorsement should be inferred.

## I. INTRODUCTION

In April 2020, children began to be hospitalized for fever and multisystem inflammation [3, 4, 5, 6], and one of the most critical clinical challenges to arise during the pandemic appeared: multisystem inflammatory syndrome in children (MIS-C). In May 2020, the Centers for Disease Control and Prevention (CDC) published a case definition for this syndrome [7], where the clinical similarity with other febrile diseases was already evident. Fever, rash, conjunctivitis, oromucosal changes, abdominal pain, vomiting, diarrhea, myocarditis, and hematological abnormalities are just some of the symptoms frequently found in MIS-C [3, 4, 5, 6, 7, 8] and that can be found in Kawasaki Disease (KD) [9], toxic shock syndrome (TSS) [10], and typhus [11, 12], generating a clinical challenge to distinguish MIS-C from these pathologies.

To address other high-level clinical challenges, computer systems designed to support the decision-making of medical teams, such as Clinical Decision Support Systems (CDSS), have been implemented. These computing systems have been implemented in a wide range of clinical challenges, such as antibiotic management [13], heart disease prediction [14], and even cancer detection [15], so the implementation of a CDSS capable of distinguishing between KD, MIS-C, typhus among other non-specific febrile illnesses would be of significant impact for medical teams in the emergency department for timely-decision making which is essential for better outcomes in these febrile conditions.

The main contributions of this paper are:

- Developed and evaluated an AI-based CDSS for distin-

guishing between KD, MIS-C, typhus, and other non-specific febrile illnesses.

- Developed and evaluated a new downsampling approach to face the problems of small imbalanced datasets.

## II. BACKGROUND

CDSSs have been used during the COVID-19 pandemic as support tools for the prognosis of disease severity [16] or predicting mortality [17], and most of them have focused only on MIS-C-related clinical challenges [18, 19]. One of the only CDSS that has extended its approach beyond MIS-C prediction is the one presented by Lam *et al.* [20], where they built a two-stage model of feedforward neural networks intended to differentiate between MIS-C, KD, and children with non-specific febrile illnesses, considering the importance of timely prediction using features obtained within the first 24 hours.

This research can be considered an extension of *Bautista-Castillo, et al.* work [21], which, to our knowledge, is the only one considering endemic typhus as one of the possible overlapping febrile diseases for MIS-C and that also incorporates a score that can be used in the Emergency Department (ED) without using electronic devices and only using features obtained during the first six hours after patient's arrival. AI-HEAT incorporates KD and children with non-specific febrile illnesses to be distinguished, being the first CDSS that considers these febrile conditions only using eight features obtained during the first six hours after Emergency Department arrival and predicting the four febrile conditions.

## III. METHODS

This section will discuss the methods used to create AI-HEAT, a CDSS based on a Triplet Loss Siamese Network that distinguishes between patients with Kawasaki, MIS-C, Typhus, and non-specific febrile illnesses.

### A. Data Imputation

Multiple Imputation by Chained Equations (MICE) [22] with LightGBM was used to address the missing values for data imputation. This is an iterative statistical technique where values are imputed several times and performed chained using LightGBM to perform the predictions in every iteration. The Python package implemented in this work can be found in [23].

### B. Cohort Creation

To avoid biases during the training and testing phases, we created four cohorts, considering three characteristics: age, sex, and the patient's condition, to obtain the most homogeneous distribution of patients possible in each cohort. This process can also be referred to as a 4-set cross-validation with matching conditions, where the matching conditions are the age, delimited in one-year intervals, the sex, and the patient's febrile condition.

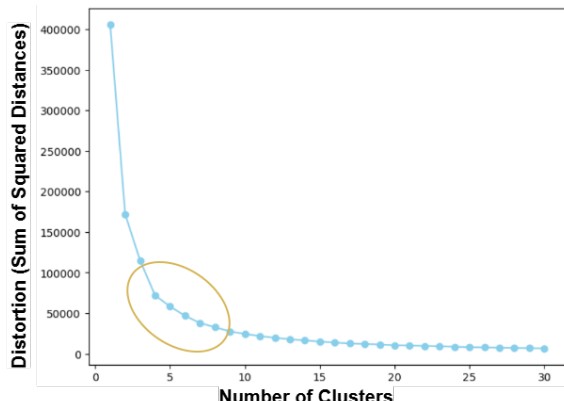

Fig. 1. Elbow method applied to get the optimal number of clusters (K) for the K-means algorithm using distortion as the metric.

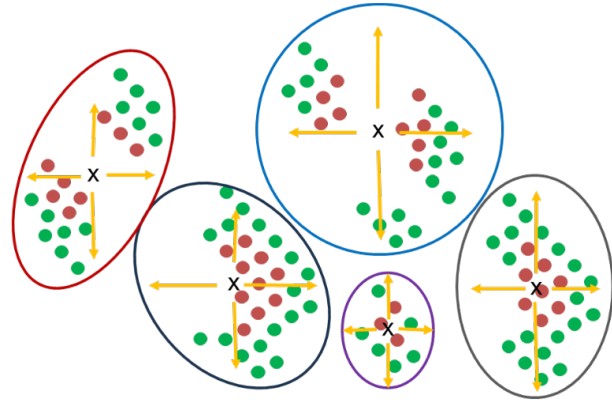

Fig. 2. Informative Samples Selection method using k-means and the Euclidean distance to select those patients that are further from the centroid because they possess the less common pattern within that cluster and are more informative to our model. All patients belong to the same majority class.

### C. Informative Samples Selection Method

To address the dataset class imbalance problem, we propose an algorithm for downsampling the majority classes. First, we perform a bi-dimensional projection of all patients from a majority class that will be used for training in the trial using t-distributed Stochastic Neighbor Embedding (t-SNE). Next, we used k-means as our clustering algorithm, defining the number of clusters (K) to find using the "elbow method," where we initialize k-means from one and iteratively augment that number until the sum of square distances, or distortion stops being considerably smaller for the next iteration, compared to the previous one to find the optimal number of clusters (Fig. 1). Finally, we compute the Euclidean distance of all points from the cluster's centroid to which they belong to determine which points are closer to the centroid and which are further from it, keeping those that are further due to they possess the less common pattern within that cluster and, therefore, they will be more informative to our model (Fig. 2). This process is iterative and is applied to all majority classes one at a time.

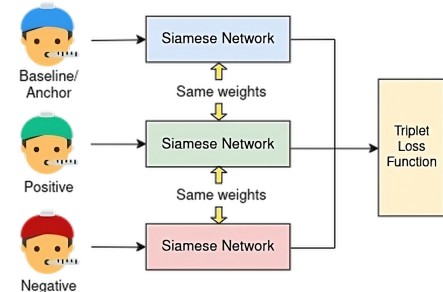

Fig. 3. Triplet Loss Siamese Network representation.

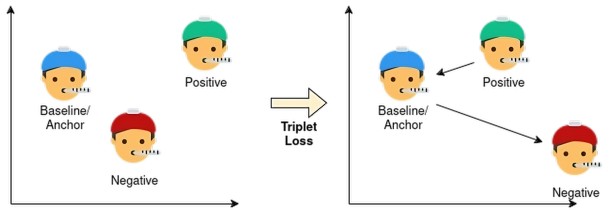

Fig. 4. Visualization of the Triplet Loss function in the embedding space.

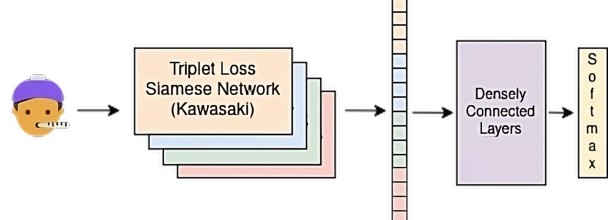

Fig. 5. AI-HEAT diagram, our deep learning model consisting of an embedding stage and a classification stage.

### D. Triplet Loss Siamese Network

The core of our deep learning model is the Triplet Loss Siamese Network. Two of the network's main characteristics are that it has three branches that share weights and require as input a positive example, a negative example, and an anchor or baseline (Fig. 3). This input is usually referred to as a triplet. This neural network aims to enforce a desired distance between the triplets that make up its input in a meaningful embedding space, reducing the distance between the anchor or baseline and the positive example and maximizing the distance between the anchor or baseline and the negative one, using the Triplet Loss function defined as:

$$\mathcal{L}(A, P, N) = max\{d(A, P) - d(A, N) + \alpha, 0\}$$

where $d(A, P)$ is the distance between the anchor or baseline and the positive example, $d(A, N)$ is the distance between the anchor or baseline and the negative example, and $\alpha$ is the minimum desired difference between the distances (Fig. 4).

### E. Deep Learning Model: AI-HEAT

AI-HEAT consists of an embedding stage and a classification stage. In the embedding stage, the patient information enters the four Siamese Triplet Loss Networks, where the representation of that patient is obtained for each of the embedding spaces in which the four febrile conditions are represented. Subsequently, the resulting embedded vectors are concatenated to form a single vector with said embedded representations. This vector will feed the classification stage, consisting of densely connected layers with a final softmax layer, whose output will be a probability distributed among the four febrile conditions given the patient's clinical and laboratory features (Fig. 5).

TABLE I
DATASET FEATURES

| ID | Feature | Description | DT |
|---|---|---|---|
| 1 | ALC : Absolute Lymphocyte Count (K/µL) | ALC laboratory test within six hours of presentation | N |
| 2 | ANC : Absolute Neutrophil Count (K/µL) | ANC laboratory test within six hours of presentation | N |
| 3 | Age (years) | Patient's age when admitted | N |
| 4 | ALT : Alanine Aminotransaminase (U/L) | ALT laboratory test within six hours of presentation | N |
| 5 | Conjunctivitis | Redness of the conjunctiva | B |
| 6 | Fever days before hospital (days) | Self-reported days of fever before admission | N |
| 7 | Rash | Abnormal change in skin color | B |
| 8 | Sex | Female = 0, Male = 1 | B |

DT=Data Type , N=Numerical, B=Binary

## IV. RESULTS

This section will discuss the characteristics related to the dataset used to train and test the AI-HEAT model and the models used as baselines, among them TabPFN [24], a prior-data fitted network that uses a transformer to classify small tabular data, XGBoost [25], Support Vector Machine (SVM), and Logistic Regression (LR), in addition to the characteristics of the cohorts and trials, the metrics used as performance assessment, and all the settings for the experimental results.

### A. Training/Testing Dataset Description

The dataset used for training and testing both models included 943 patients admitted with non-specific febrile illnesses and 1,105 patients admitted with KD to Rady Children's Hospital and its satellite locations, 135 patients admitted with MIS-C and 87 patients admitted with murine typhus admitted to Texas Children's Hospital and its two satellite campuses within the greater Houston area. Medical records were reviewed, with eight demographic, clinical, and laboratory features available within six hours of the presentation for all febrile conditions (Table I). Within the dataset, 1,347 patients are males, and 923 are females. Maximum, minimum, mean, prevalence, and missing values for all the features of the dataset are shown in Table II.

### B. Trials

After applying the 4-set cross-validation with matching conditions to create the four cohorts for training and testing, the distributions shown in Table III were obtained. Once the

as follows:

TABLE II
DATASET STATISTICS

| Feature | Numerical | | | Binary Prevalence | Missing |
|---|---|---|---|---|---|
| | Min | Max | Median | | |
| ALC | 0.07 | 17.42 | 2.52 | - | 96 |
| ANC | 0.38 | 37.97 | 6.99 | - | 92 |
| Age (years) | <1 | 19 | 3 | - | 0 |
| ALT | 3 | 1,045 | 33 | - | 273 |
| Conjunctivitis | - | - | - | Yes: 71% No: 29% | 5 |
| Fever days before hospital (days) | 0 | 15 | 5 | - | 1 |
| Rash | - | - | - | Yes: 81% No: 19% | 5 |
| Sex | - | - | - | Female: 41% Male: 59% | 0 |

TABLE III
NUMBER OF PATIENTS IN EACH COHORT

| | Cohort 1 | Cohort 2 | Cohort 3 | Cohort 4 | % of Total |
|---|---|---|---|---|---|
| Febrile Control | 250 | 241 | 232 | 220 | 42 |
| Kawasaki | 292 | 278 | 271 | 264 | 48 |
| MIS-C | 50 | 38 | 28 | 19 | 6 |
| Typhus | 34 | 24 | 18 | 11 | 4 |
| Total | 626 | 581 | 549 | 514 | 100 |

cohorts were created, we defined the training and testing patients for each trial based on these cohorts, where for each trial, the cohort with the same Trial number will be used as a test cohort. For Trial 1, the cohort used as a test set was Cohort 1; for Trial 2, the cohort used as the test was Cohort 2, and so on. Finally, for the cohorts used as training for each trial, the Informative Samples Selection Method (Sec. III-C) was applied, obtaining the distribution of the febrile conditions for training for every trial shown in Table IV.

### C. Performance Assessment

The Balanced Accuracy metric was used to evaluate the performance of AI-HEAT and the baseline models. This decision was made based on the evident imbalance of the cohort used for testing in each trial, which is not subject to the Informative Sample Selection Method and thus retains the characteristics shown in Table III. The Balanced Accuracy metric was defined

TABLE IV
DISTRIBUTION OF PATIENTS FOR EACH FEBRILE CONDITION ACROSS ALL TRIALS

| | Trial 1 | Trial 2 | Trial 3 | Trial 4 | % of Total |
|---|---|---|---|---|---|
| Febrile Control | 67 | 79 | 90 | 98 | 24 |
| Kawasaki | 78 | 92 | 103 | 111 | 28 |
| MIS-C | 85 | 97 | 107 | 116 | 29 |
| Typhus | 53 | 63 | 69 | 76 | 19 |
| Total | 283 | 331 | 369 | 401 | 100 |

$$\text{Balanced Accuracy} = \frac{1}{N} \sum_{i=1}^{N} \text{Sensitivity}_i,$$

where $N$ is the number of febrile conditions.

### D. Experimental Results

The training and testing of all models were performed with Python 3.9.17, Tensorflow 2.13.0, Pandas 2.0.3, and Keras 2.13.1 running on a LINUX-based computer equipped with an AMD Ryzen 5 5600g CPU and a NVIDIA GeForce RTX3060 GPU. For TabPFN, we used the Python package downloaded directly from the authors' GitHub repository [26]. Similarly, XGBoost was installed and ran following the steps outlined in the developers' documentation [27]. SVM and LR were implemented directly from the scikit-learn package for Python [28]. To train AI-HEAT, our CDSS, we began by training each Triplet Loss Siamese network. These networks were designed to create an embedding space for each of the four febrile conditions. In this setup, each febrile condition was treated as a positive instance within its embedding space, while the other conditions were considered negative. Once the four Triplet Loss Siamese Networks had been trained, their outputs were fed into a concatenation layer that combines the vectors from each network and pass them through a series of densely connected layers that form the classification stage of our CDSS. Each Triplet Loss Siamese Network consists of four layers with 16, 32, 16, and 8 neurons, respectively. Training was performed using the Triplet Loss function with a learning rate of 0.002, an alpha parameter of 0.2, and a batch size of 30. The classification module uses three densely connected layers, with 32, 64, and 32 neurons, respectively, and employs a ReLU activation function. For training this module, we used mean square error as the loss function and an Adam optimizer with a learning rate of 0.001 over 25 epochs, with ten steps per epoch. The training was conducted for all models with and without the Informative Sample Selection method implementation to compare the impact of the downsampling method proposed in III-C. In the case of TabPFN, due to the Transformer's restrictions that do not admit more than 1024 samples for training, a random sampling of 400 samples was carried out for non-specific febrile illnesses and KD patients. The experimental results for all trials are shown in Table V, while experimental results for all classes can be seen in Fig. 6.

### V. DISCUSSION

In the results shown in Table V and Figure 6, the positive impact of implementing the Informative Samples Selection Method is evident, where the models most benefited by its implementation were SVM and AI-HEAT, and the least benefited was TabPFN. This may be due to TabPFN's well-known robustness even when dealing with imbalanced datasets thanks to its Transformer-type architecture. AI-HEAT achieves a performance similar to that of the TabPFN, with some specific advantages such as being a model that allows visualization

TABLE V
BALANCED ACCURACY RESULTS FOR ALL TRIALS

| | Imbalanced | | | | | Balanced | | | | |
|---|---|---|---|---|---|---|---|---|---|---|
| | **LR** | **SVM** | **XGBoost** | **TabPFN** | **AI-HEAT** | **LR** | **SVM** | **XGBoost** | **TabPFN** | **AI-HEAT** |
| **Trial 1** | 0.51 | 0.48 | 0.55 | 0.64 | 0.50 | 0.64 | 0.63 | 0.61 | 0.66 | **0.69** |
| **Trial 2** | 0.49 | 0.42 | 0.53 | 0.65 | 0.54 | 0.68 | 0.65 | 0.66 | **0.75** | 0.72 |
| **Trial 3** | 0.52 | 0.47 | 0.64 | 0.66 | 0.51 | 0.69 | 0.70 | 0.70 | **0.78** | 0.74 |
| **Trial 4** | 0.55 | 0.49 | 0.69 | 0.69 | 0.54 | 0.69 | 0.73 | 0.73 | 0.75 | **0.78** |
| **Average** | 0.52 | 0.47 | 0.60 | 0.66 | 0.52 | 0.68 | 0.68 | 0.67 | **0.74** | 0.73 |

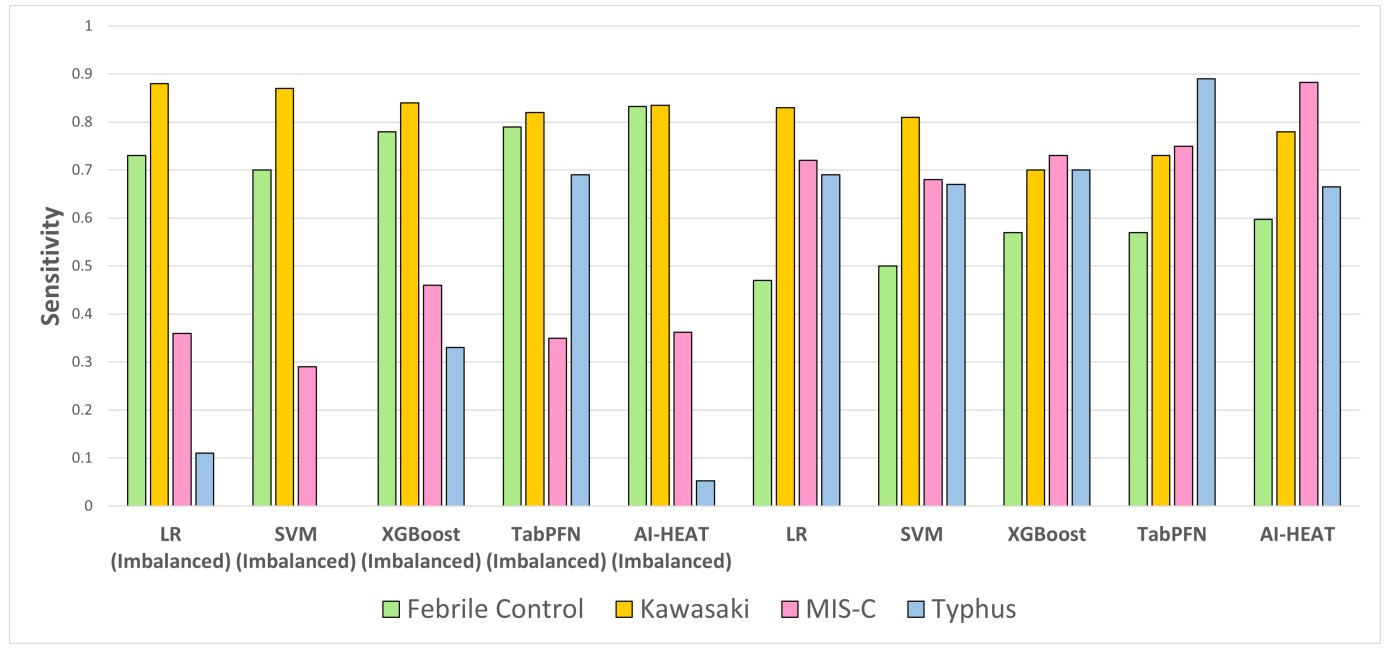

Fig. 6. Experimental results per class for every baseline model and AI-HEAT before and after Informative Samples Selection Method.

of the embedding spaces (Fig. 7, 8, 9, and 10) that can result in a better understanding of the similarity of some of the febrile conditions treated here, as well as the significant difference in size of both architectures, where TabPFN has 25.8 M parameters. In comparison, AI-HEAT only has 24.4 K parameters.

In these bidimensional projections of the multi-embedding space, it can be observed how AI-HEAT interprets the existence of a certain similarity in febrile conditions such as Kawasaki (Fig. 8) and typhus (Fig. 10), where it observes that several patients occupy the same regions of this multi-embedding space. At the same time, it can be seen that patients with MIS-C (Fig. 9) are distributed throughout the multi-embedding space, which suggests that they are the most challenging patients to distinguish between the four febrile conditions. These conclusions were confirmed by medical staff who are in constant contact with these febrile conditions.

## VI. CONCLUSIONS

AI-HEAT is a CDSS capable of obtaining performance similar to that of a state-of-the-art Transformer-type deep learning model such as TabPFN with advantages such as being almost a thousand times smaller, having the ability to

allow the visualization of its embedding spaces to the better understanding of the febrile conditions that it distinguishes, and having the flexibility to be able to incorporate other architectures in its classification stage that can improve its performance.

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
