# OpenReview forum: "AI-HEAT: A Clinical Decision Support System for pediatrics febrile conditions"
_IEEE.org/EMBS/BHI/2024/Conference — IEEE BHI'24_

### Official Review · Reviewer_qoe8 · 2024-08-08
**AI for febrile conditions in children**

**Overall Rating:** 7
**Confidence:** 4

**Other Quality Metrics:**

a) great
b) great
c) fair
d) good

**Questions For The Authors:**

By using cross-validation, you may have more precise results by running the training (and validation) 4 times instead of 1, averaging out the results. This avoids accidental over-training and other dataset statistical anomalies which may affect the training process.

In algorithm 1, are age intervals and sex considered "condition"? this might be confusing.

Minor notes: all line numbers for Algorithm 1 are "0" which defeats the purpose of having line numbers, they are not referred later in the paper, so maybe it would be good to remove them. In  IV. B. a reference to Algorithm A is stated, probably this is Algorithm 1.

**Strengths:**

Clear structure, scientific rigour, excellent english aiding comprehensibility, comparison with other similar systems.

**Summary Of The Paper:**

A solution based on embeddings of patient parameters and neural networks is described, parametrized, evaluated and compared with SotA systems specialising in febrile conditions in children

**Weaknesses:**

Algorithm 1 (later referred as Algorithm A) might just be a 4 set cross-validation segmentation of the dataset with matching conditions.

---

### Official Review · Reviewer_Ni7G · 2024-08-16
**Review of AI-HEAT, an AI-based CDSS to distinguish challenging febrile illnesses**

**Overall Rating:** 6
**Confidence:** 3

**Other Quality Metrics:**

Clarity of writing: fair
Clinical Significance: fair
Methodological Novelty: fair
Experiments and Results: good

**Questions For The Authors:**

NA

**Strengths:**

AI-HEAT has comparable performance with state-of-the-art methods when using balanced dataset. Also, due to its modular framework design, it can visualize its embedding space which can be beneficial.

**Summary Of The Paper:**

In this paper, authors present AI-HEAT, a clinical decision support system (CDSS), to facilitate distinguishing febrile illnesses with similar symptoms that makes their diagnosis challenging. AI-HEAT consists of two modules; an embedding module, where patient information is encoded via a Triplet Loss Siamese Network, and a classification module where this encoded information is fed to a fully connected neural network. Furthermore, a sample selection method is utilized to tackle issues regarding imbalance datasets, which its importance is demonstrated in results.

**Weaknesses:**

From the technical point of view, although AI-HEAT has comparable performance as state-of-the-art methods, its performance is very sensitive to the dataset, which can be unacceptable in medical scenario. Specifically, when using imbalance dataset, basically there is no advantage of using AI-HEAT. The proposed sampling method is not necessarily new (margin-based sampling, in Active Learning literature). In general, paper lacks originality and does not present any significant novel contributions.

From the paper structure point of view, the paper is not properly organized, and the manuscript can be improved.  In different sections, like in IV.C, and at the end of IV.D, the text is poorly written and is ambiguous. Most figures have very low quality, specifically figures 3, 4, and 5. It is much more helpful to put all 4 figures regarding the visualizations in one figure with 4 subfigures. In section IV.A, in table I it is mentioned that there are 4 binary features, however, statistics of two of them is reported in table II. Also “Fever days before hospital” is denoted as binary in table I, while in table II it is considered as numerical. The abbreviations under table I are better to be mentioned in the manuscript or with a more distance from table. In the explanation of table II, authors mention "prevalence" however there is no such term in table II. Clearer labels are required as well (figure 6 does not have any label regarding the values).

---

### Decision · Program_Chairs · 2024-09-23

Accept